# Handrim Reaction Force and Moment Assessment Using a Minimal IMU Configuration and Non-Linear Modeling Approach during Manual Wheelchair Propulsion

**DOI:** 10.3390/s24196307

**Published:** 2024-09-29

**Authors:** Rachid Aissaoui, Amaury De Lutiis, Aiman Feghoul, Félix Chénier

**Affiliations:** 1Laboratoire de Recherche en Innovation Ouverte en Technologie de la Santé, Centre de Recherche CRCHUM, Montreal, QC H2X 0A9, Canada; delutiis.amaury@gmail.com (A.D.L.); aimanfeghoul@gmail.com (A.F.); 2Département de Génie des Systèmes, École de Technologie Supérieure, Montréal, QC H3C 1K3, Canada; 3Centre de Recherche Interdisciplinaire de Réadaptation de Montréal, Montreal, QC H3S 1M9, Canada; chenier.felix@uqam.ca; 4Regroupement Scientifique INTER, Technologies Interactives En Réadaptation, Sherbrooke, QC J1K 0A5, Canada; 5Département des Sciences de l’activité Physique, Université du Québec à Montréal, Montréal, QC H2X 1Y4, Canada

**Keywords:** inertial measurement unit, manual wheelchair, handrim reaction forces and moments, Hammerstein–Wiener model, recurrent neural network BiLSTM

## Abstract

Manual wheelchair propulsion represents a repetitive and constraining task, which leads mainly to the development of joint injury in spinal cord-injured people. One of the main reasons is the load sustained by the shoulder joint during the propulsion cycle. Moreover, the load at the shoulder joint is highly correlated with the force and moment acting at the handrim level. The main objective of this study is related to the estimation of handrim reactions forces and moments during wheelchair propulsion using only a single inertial measurement unit per hand. Two approaches are proposed here: Firstly, a method of identification of a non-linear transfer function based on the Hammerstein–Wiener (HW) modeling approach was used. The latter represents a typical multi-input single output in a system engineering modeling approach. Secondly, a specific variant of recurrent neural network called BiLSTM is proposed to predict the time-series data of force and moments at the handrim level. Eleven subjects participated in this study in a linear propulsion protocol, while the forces and moments were measured by a dynamic platform. The two input signals were the linear acceleration as well the angular velocity of the wrist joint. The horizontal, vertical and sagittal moments were estimated by the two approaches. The mean average error (MAE) shows a value of 6.10 N and 4.30 N for the horizontal force for BiLSTM and HW, respectively. The results for the vertical direction show a MAE of 5.91 N and 7.59 N for BiLSTM and HW, respectively. Finally, the MAE for the sagittal moment varies from 0.96 Nm (BiLSTM) to 1.09 Nm for the HW model. The approaches seem similar with respect to the MAE and can be considered accurate knowing that the order of magnitude of the uncertainties of the dynamic platform was reported to be 2.2 N for the horizontal and vertical forces and 2.24 Nm for the sagittal moments. However, it should be noted that HW necessitates the knowledge of the average force and patterns of each subject, whereas the BiLSTM method do not involve the average patterns, which shows its superiority for time-series data prediction. The results provided in this study show the possibility of measuring dynamic forces acting at the handrim level during wheelchair manual propulsion in ecological environments.

## 1. Introduction

Manual wheelchair propulsion represents a repetitive and constraining task, which leads to the development of joint injury in spinal cord-injured people. The shoulder joint remains the site where injury and pain are the most prevalent [1]. The net compressive force developed at the shoulder joint is related to a surrogate variable, which is generally represented by the resultant shoulder joint moment [2]. The latter is generally obtained by a recursive inverse dynamic method [3,4]. To be able to estimate the shoulder moment, the forces and the moment reactions at the handrim level should be measured [5]. It has been shown in the past that an increase in the shoulder joint moment is related to the pattern of handrim forces and moment reactions [6,7]. In a laboratory set-up, it is relatively easier to measure handrim reaction forces when using a haptic wheelchair simulator and dynamic force platforms [6]. However, outdoor force measurement remains impossible to estimate when using the patient’s wheelchair, i.e., a non-instrumented wheel. Moreover, even when using instrumented wheels such as the Smartwheel, the inertia characteristic of the wheels as well as their frontal plane orientation modify the intrinsic configuration of the user’s seating. This will influence the handrim reaction forces and moments.

The assessment of ground reaction forces during walking and running gait has been tackled for many years now, using the inertial measurement unit (IMU). In fact, a systematic review has been reported spreading over two decades for the estimation of ground reaction forces [8]. There are two major approaches to the problem: a multibody approach and local segment approach. The multibody approach is generally based on inverse dynamics as applied to the all-body segment, i.e., the Newton–Euler formulation. The total vertical ground reaction forces are equal to the sum of the mass of each segment multiplied by the vertical linear acceleration of the segment minus the gravitation vector. This method is ideal for a perfect multibody model, but not in human model segments. The global approach suffers from three issues: the first one is related to the skin tissue artifact of markers or sensors, the second one is related to the estimation of the location and mass of each segment of the body, and the third one is more complex and related to the smooth transition in the double-support phase. The study conducted by the authors of [9] reported an RMSE of almost 64 N for the vertical direction and 43 N for the horizontal direction, as well as 18 Nm for the sagittal moment during normal walking. Recently, a probabilistic approach based on principal component analysis has been applied to the distribution of ground reaction forces between legs and reduced the error on vertical ground reaction forces to 2.5 N/kg [10]. Moreover, when there is multiple contact, such as in lifting a box, the multi-body approach becomes inefficient and necessitates an optimization approach as well as modeling of the contact between the environment and the human body; the latter can be handled by a complementary approach [11]. The latter estimated the vertical reaction forces with an RMSE of 0.51 N/kg.

The local approach is a method which tries to relate the information localized in one part of the body to the forces acting at the ground level. In [12], one accelerometer was fixed at the hip level, and a logarithmic regression equation was developed to predict peak ground reaction forces. The RMSE was large and almost close to 150 N. Recently, the local approach based on three uniaxial load cells fixed on the shoes and combined with a deep learning method (Long Short-Term Memory) was used to estimate the ground reaction forces [13]. The latter provide an RMSE of 65 N in the vertical direction and 15 N in the horizontal direction. The advantage of the local approach is definitively the number of sensors used. Moreover, it seems that the closer the sensors to the contact zone, the better the estimated GRF.

In the field of manual wheelchair propulsion, few attempts have been made in the past to estimate the handrim reaction forces. A four-bar linkage was developed to model upper-body movement using an optimization approach to minimize the shoulder moment as well as to deduce the magnitude and orientation of sagittal handrim reaction forces [14]. In [14], the relative error in estimating the ground reaction forces varies from 16 to 88% during the push cycle. In [15], a kinematic method was designed to estimate the average global push force using an impulse–momentum equality constraint during the wheelchair propulsion cycle. The authors found a relative error that varies between 31% and 43% at the comfortable and maximum linear velocities [15]. In [16], the authors used 36 feature variables that represent the angular displacement, velocity and acceleration of the shoulder, elbow and wrist joints as well as anthropometric data as an input to a recurrent neural network in order to predict the magnitude and the orientation of the handrim reaction forces. The relative error on magnitude varies during the push cycle from 86% at the beginning of the push cycle to 13% at the peak force level. It should be noted that the two studies use an optoelectronic measurement system to measure the kinematics of the upper body during manual wheelchair propulsion. These methods are considered multibody methods and have two major drawbacks: first, in wheelchairs, multiple contact exists between the user and the seat interface, which precludes any use of a regular Newton law as in gait analysis [17]; secondly, the methods are inapplicable outside the laboratory, since in general they necessitate a stationary ergometer. Moreover, none of these methods estimates the bilateral sagittal handrim moment of propulsion. The purpose of this study is to develop a new ambulatory local approach that uses one inertial measurement unit (IMU) sensor at each wrist level. The IMU can provide two inputs: the 3D linear acceleration and 3D angular velocity vectors. Traditional forecasting of time-series data used a block-oriented identification method such as ARIMA [18], which represents a linear approach and performs poorly when there is a non-linear relationship between input and output time series. A non-linear identification modeling approach such as Hammerstein–Wiener (HW) exists and has been used in the past to forecast time-series data and prove its superiority to ARIMA; however, it has never been used in wheelchair biomechanics [19]. The purpose of this study is to compare two approaches, namely, HW and BiLTSM, in the forecasting of the time series of reaction forces and moments acting at the handrim level during manual wheelchair propulsion.

## 2. Materials and Methods

### 2.1. Experimental Set-Up

Eleven healthy young subjects participated in this study. Their mean age, weight and height were equal to 27.3 (4.9) years, 73.0 (13.4) kg and 174.0 (7.3) cm. They were asked to propel a manual wheelchair (Ultralight Action A4, Invacare) at 1 m/s along 20 m in a linear direction towards a hallway (Figure 1). All subjects signed a consent form, which was approved by the ethical committee of the École de technologie supérieure (H20150508).

The wheelchair was equipped with two Smartwheels which measured the handrim reaction forces on the anteroposterior and vertical directions, as well as the moment reaction on the medial–lateral axis. The sampling frequency was fixed at 240 Hz. The Smartwheel software filtered the raw force and moment data with a 2nd-order low-pass Butterworth filter with a cut-off frequency of 30 Hz. During the propulsion, the Xsens System (Mvn Biomech, Xsens Inc., Henderson, NV, USA) was used to model the head, trunk and upper limb of the body, represented by 10 IMU sensors. A calibration procedure in sitting position, i.e., a T-pose, was made for each subject. The IMU data were first processed by the Xsens system (i.e., internal filtering algorithm using Kalman filter). Moreover, gravity was removed by the Xsens system following the estimation of the quaternion of each sensor in the reference system of the Xsens. The Xsens has a biomechanical model which aligns the sensor information with the human body. In this study, we use the only the norm of the linear acceleration and angular velocity vector for the left and right wrist joints. The sampling frequency of the Xsens was set at 120 Hz. A numerical synchronization was established by asking the subject to kick the handrim with their right hand at the beginning and the end of trial. These two events (Tc1 and Tc2) are easily distinguishable in the time series of force and hand acceleration (Figure 2).

### 2.2. Data Processing

A resampling method using the <resample> time-series function from MATLAB (R2019b) was used to down sample and synchronize the force and moment from the Smartwheel with respect to the IMU data without filtering the data. The sagittal moment Mz of the right and left sides was used to define the push cycle and to normalize the time of propulsion from 0 to 100%. However, for the identification process, a 20% extra signal for each side of the normalized push cycle was also used. In this case, the total length of each cycle was around 140%. For each push cycle, we normalized the magnitude of the moment and forces acting at the handrim, as measured by the Smartwheel, by their respective rms (root mean square) cycle value; this helps to keep the moment Mz fluctuating around an average value of 2.5 during wheelchair propulsion. This normalization technique is more consistent that the one using maximal value in each cycle.

### 2.3. Non-Linear Hammerstein–Wiener Modeling Approach

In each trial, between 30 and 38 steady-state push cycles were obtained for each subject. The HW model consist of 3 time-series inputs, namely, the norm of the linear acceleration of the wrist joint, the norm of the angular velocity of the wrist joint as well as the average normalized pattern of the force/moment. These 3-time series are considered as an input to the HW approach, whereas the output is the actual measured normalized force or moment. Figure 3 shows a SISO system, which consists of 3 blocs (NLN), i.e., non-linear followed by a linear and followed again by a non-linear bloc. Meanwhile, Figure 4 shows a MISO approach, i.e., a combination of three SISO systems in one MISO system which consists of 4 non-linear blocs and 3 linear blocs. There are a variety of possible non-linear blocs [20,21]. In this study, piece-wise non-linear blocs of the 4th order were chosen. This means that each non-linear bloc has 4 parameters to be determined. The linear blocs consisted of number of poles and zeros which represent the transfer function in the z transform domain. In this study, for each linear bloc we vary the number of poles from 1 to 6 and consequently the numbers of zeros from 1 to *n* − 1, where n is the number of poles. Fifteen cycles of one subject were used to train and identify the Hammerstein–Wiener model in order to define the best architecture of the linear blocs’ transfer function using the prediction error minimization, and the rest of the cycles (thirty) were used to test the prediction of the model.

For each number of poles varying from *n* = 6 to 1 there corresponds a number of zeros from *n* − 1 to 1. In this case, for example, the predicted moment Mz was assessed using 9261 combination models for the identification process. From this, we kept the 111 combination models of poles and zeros that gave the best results. The total number of non-linear parameters is 16 (4 blocs with 4 orders) and the total of number of linear parameters is higher, at 33 (i.e., 3 * (6poles + 5 zeros). Since there was a lot of combination, we repeated this process until we found the best combination for our typical subject. As an example, the following linear transfer function was found for the moment Mz output, where *H*_1_(*z*) (2 zeros, 5 poles) corresponds to the input linear acceleration, *H*_2_(*z*) (5 zeros, 6 poles) corresponds to the angular velocity and *H*_3_(*z*) (2 zeros, 5 poles) corresponds to the generic normalized pattern of Mz:$$H_{1}\left(z\right)=\frac{-0.60\left(1-0.86z^{-1}\right)\left(1-0.86z^{-1}\right)}{\left(1+0.83z^{-1}\right)\left(1+0.22z^{-1}\right)\left(1-0.09z^{-1}\right)\left(1-1.97z^{-1}+0.97z^{-2}\right)}$$
$$H_{2}\left(z\right)=\frac{0.11\left(1-0.98z^{-1}\right)\left(1-1.99z^{-1}+0.99z^{-2}\right)\left(1-1.83z^{-1}+0.85z^{-2}\right)}{\left(1-0.33z^{-1}\right)\left(1-0.15z^{-1}\right)\left(1-1.96z^{-1}+0.98z^{-2}\right)\left(1-1.99z^{-1}+z^{-2}\right)}$$
$$H_{3}\left(z\right)=\frac{-0.50\left(1-1.99z^{-1}+1.00z^{-2}\right)\left(1-1.83z^{-1}+0.85z^{-2}\right)}{\left(1-0.36z^{-1}\right)\left(1-1.99z^{-1}+0.99z^{-2}\right)\left(1+0.54z^{-1}+0.63z^{-2}\right)}$$

Knowing the number of poles and zeros for the best linear combination, this architecture was applied for each subject. Each combination takes 167.64 s to process. It took almost 18 days to process all of the models.

### 2.4. Neural Network Long-Short Term Memory (LSTM) Approach

After using the H-W model, we decided to compare it with a deep learning method. The recurrent neural network “Long Short-Term Memory”, considered most suited for time-series data, was chosen [22]. The BiLSTM neural network that goes both forward and backward through data gave better result than simple LSTM to predict the handrim moments and forces [23]. The input data consist of only 2 time series, namely, the norm of linear acceleration and the norm of angular acceleration, and the output is the corresponding force or moment measured time series.

In this method, one subject was used as the test subject each time (i.e., leave one out). The rest of the data were separated between training and validation data in an 80% to 20% ratio. This separation was made to maintain maximum peak homogeneity between training and validation. The limited data of 11 subjects made the use of all subjects for the validation of data ineffective in our study. The validation of data was used to make sure the training process was successful by comparing the RMSE and Loss of the training and validation data. Figure 5 shows a successful training with both training and validation data.

When compared to the Hammerstein–Wiener modeling approach, the BiLSTM model did not need any specific pattern as input and used only the norm of the linear acceleration vector and the norm of the angular velocity vector as input. These inputs were normalized with the same RMS method as in the H-W model. Figure 6 shows the complete recurrent neural network model. The “Sequence Input Layer” adapts input data for 3 BiLSTM layers of 400 hidden units followed by a fully connected layer to predict our data.

The training process took almost 16 min with the following computer configuration: an Intel Core vPro i9 processer associated with an Nvidia RTX A4000 GPU graphic card. The process adjusted the layers to have the optimal prediction. The training process had 300 epochs that provided a high performance/speed ratio in training, meaning that the training uses all the data 300 times. This model was trained 13 times, leaving each subject out of training once with the LOSO method. The measured and the predicted outputs were finally compared together in each test.

### 2.5. Statistical Analysis

To compare the predicted value with respect to the measured one, the RMS value of the predicted and measured variables was assessed. A Bland–Altman graphical figure was computed to assess the limit of agreement between the predicted and the measured variables [24]. Moreover, an analysis of variance (Anova) was conducted on the peak level on the right and left sides for each dependent variable Fx, Fy and Mz to test the statistical difference between the measured and the predicted variables. The MAE and RMSE values were also assessed.

## 3. Results

### 3.1. Non-Linear Hammerstein–Wiener Modeling Approach

Figure 7 exhibits typical input time-series data for the linear acceleration and angular velocity of the right hand of a typical subject (S008) during the manual wheelchair propulsion cycle. The timing of Figure 7 represents 38 cycles, and the average value for the linear acceleration of the right hand was equal to 7.89 (+/−5.12) m/s^2^, whereas the maximal value reached 52.13 m/s^2^. The mean average angular velocity reaches the value of 2.98 (+/−1.18) rad/s. The maximal value of angular velocity was equal to 9.75 rad/s.

Figure 8 shows the results obtained for a typical subject (S008) for the right and left sides. The predicted and the measured time-series signals follow in general the same pattern using the H-W modeling approach. It is noteworthy that even the 20% periods that precede and follow the push phase have been well predicted.

Table 1 shows the results of the prediction throughout the continuous cycle for the 11 subjects. It shows for each dependent variable the RMSE as well as the MAE for the left and right sides. For the Fx component, the mean RMSE varies from 5.6 to 5.7 N for left and right side, respectively. The mean MAE for FX is lower (4.4 and 4.2 N). The vertical force direction exhibits a slightly higher value for the RMSE (9.4 and 9.7 N), and 7.5 to 7.6 N for the MAE. The sagittal moment exhibits lower value for RMSE and MAE also (1 to 1.1 Nm). In general, the right and left side look similar with regard to the prediction accuracy.

Figure 9 shows the Bland–Altman graph for the peak value in the horizontal force direction. All of the subjects are within the limit of agreement. However, we note a small but a significant difference in bias for the right-hand propulsion.

Figure 10 shows the results of the vertical direction and that all subjects belonged to the interval of the limit of agreement with a non-significant bias.

The Bland–Altman graph in Figure 11 shows that all of the subjects (except one) belonged to the interval of limit of agreement with no significant differences in the bias between the measured and predicted sagittal moments.

### 3.2. Recurrent Neural Network: Bi-Long Short-Term Memory (BiLSTM) Approach

Figure 12 shows the mean result obtain for the BiLSTM prediction of all subjects on the right and left sides. The pattern of the measured and predicted time series follows the same path. There was even a good prediction before and after the push phase.

Table 2 shows the average value of RMSE and MAE of the BiLSTM prediction for twelve subjects. The average RMSE of the horizontal force on the right side varies from 4.4 N to 14.0 N and the MAE from 3.6 N to 11.2 N. The mean RMSE and MAE are, respectively, 7.4 N and 6.1 N. On the left side, the RMSE varies from 3.7 N to 11.4 N and the MAE from 3.0 N to 10.9 N. The mean RMSE and MAE are, respectively, 7.3 N and 6.1 N. The difference between the mean RMSE and MAE on each side is close zero, so the prediction is evenly effective on the right and left sides. For the vertical force, the average RMSE on the right side varies from 5.1 N to 10.1 N and the MAE from 4.1 N to 8.0 N. The mean RMSE and MAE are 7.5 N and 5.8 N. On the left side, the RMSE varies from 5.4 N to 10.9 N and the MAE from 4.6 N to 8.8 N. The mean RMSE and MAE are, respectively, 7.6 N and 5.9 N. The sagittal moment has an average RMSE value on the right side which varies between 0.9 N.m and 1.6 N.m and the MAE between 0.7 N.m and 1.2 N.m. On the left side, the RMSE varies between 0.9 N.m and 1.5 N.m and the MAE between 0.9 N.m and 1.5 N.m. On both sides, the mean RMSE is 1.2 N.m and the mean MAE is 0.99 N.m. No difference was shown in the mean value errors between sides.

For all forces and moments, the difference in the mean RMSE and MAE on the right and left sides are very close, so the prediction of these forces and moments are evenly effective on each side.

## 4. Discussion

The purpose of this paper was to predict handrim reaction forces and moments during linear wheelchair propulsion using only one IMU sensor per hand. It is noteworthy to see that there are few studies that concern this issue in comparison to the standard form of gait locomotion. Our results indicate that the methods proposed here, i.e., the identification of the process and the use of BiLSTM, have a high accuracy when compared to earlier studies [9,11]. In fact, the identification method shows a mean average error (MAE) in the horizontal direction of around 4.5 and 5.7 N for left and right handrims, respectively, 0.2 N and 1.7 N smaller than the BiLSTM model. The vertical direction has a slightly higher value, i.e., 9.0 and 9.3 N for left and right sides (identification process). This represents values 3.4 N and 4.2 N higher than the MAE obtained with the BiLSTM model. In terms of the sagittal moment acting at the handrim hub, the MAE was 1.1 and 1.2 N.m for left and right sides, respectively, 0.2 N.m and 0.1 N.m higher than that of the BiLSTM model. However, to the knowledge of the authors, no data have predicted the sagittal moment either in study [14] or study [16]. BiLSTM has better prediction for data with less variability in their pattern, like the sagittal moment and the vertical direction.

The statistical analysis in our studies was based merely on Bland–Altman analysis as well as Anova analysis. For the BiLSTM model, the horizontal direction revealed a significant negative bias at the peak level of 2.2 N (*p* < 0.01) with respect to the Smartwheel. Meanwhile, for the right side the negative bias of 2.4 N (*p* > 0.05) was found non-significant. The value of the bias is considered small and negligible in this study since the peak value in the horizontal direction can reach 60 N. The vertical direction reveals some bias of 2.0 N and 0.35 N for the right and the left sides at the peak value level. These two biases, however, were found statistically non-significant. During the linear propulsion, the MAE of the peak moment was found to be 1.1 to 1.2 N.m for the right and left levels. The bias found using the Bland–Altman graph was equal to 0.31 N.m and 0.15 N.m for the right and left sides. These biases were found non-significant. In general, and for all the subjects except one, the results indicate an agreement between the measured and the predicted force and moment, since the limit of agreement is larger and includes all of the data. Our bias is lower than the one found in study [25], in which the agreement between the Smartwheel instrumentation and the instrumented split-belt treadmill was around 0.51 N.m for the average moment and 0.71 N.m for the peak moment.

Our method has the advantages of minimal configuration, i.e., it necessitates only two IMU sensors that can be affixed onto the hand or wrist joint. The fact that the earlier methods were not be able to reach high accuracy resides in the fact that multibody systems with many contacts such as a seat, backrest and legrest are more difficult to model in practice [17]. Our proposal is to be close to the contact level of the handrim, i.e., hand or wrist joint. In this way, the dynamic system of order six was sufficient to find a non-linear transfer function between three inputs, namely, the linear resultant acceleration of the hand, the angular velocity resultant as well as the normalized pattern of the forces or the moment, and the output are the forces and the sagittal moment. The purpose of the normalized generic pattern of the force and the moment is that it has a modulator effect on the actual kinematic input by the subject. Moreover, our method necessitates only 15 push cycles to identify (i.e., to learn) the dynamic process. The linear and non-linear functions found by the H-W approach are actually used a at the right and left sides and for all the subjects. The same transfer function was able to be applied to all subjects. On the other hand, our study also shows that the recurrent neural network BiLSTM is a great alternative to the non-linear transfer function method and does not need the generic normalized pattern to be performed. The method to use can be chosen depending on the variability between the predicted patterns.

This study, however, is limited to straightforward propulsion, i.e., in the linear direction; we did not assess a curvi-linear or slope path. Moreover, the wheel camber in this study was fixed to zero, i.e., the inplane of the smartwheel is the vertical plane. In many wheelchairs, such as those dedicated to spinal cord injury and for sport activity, the camber is important. The subjects that participated in this study were young and healthy, i.e., non-wheelchair users [26]. The generalization of this method to wheelchair users such as those with spinal cord injury should be carried out in a future study. Another aspect about the normalization used in this study is that the rms of the cycle was used as a base and the use of this base explicitly for the HW method is considered a limitation. The generic pattern is also a limitation of the HW method. Future work should be carried out to understand how to deformalize the time-series signal after the prediction process.

In conclusion, this paper presents two methods based on a block-oriented non-linear identification process, namely the Hammerstein–Wiener modeling approach and a recurrent neural network based on the BiLSTM architecture. Both methods provide accurate results; however, the HW method, for instance, necessitates knowledge of the generic pattern.

## Figures and Tables

**Figure 1 sensors-24-06307-f001:**
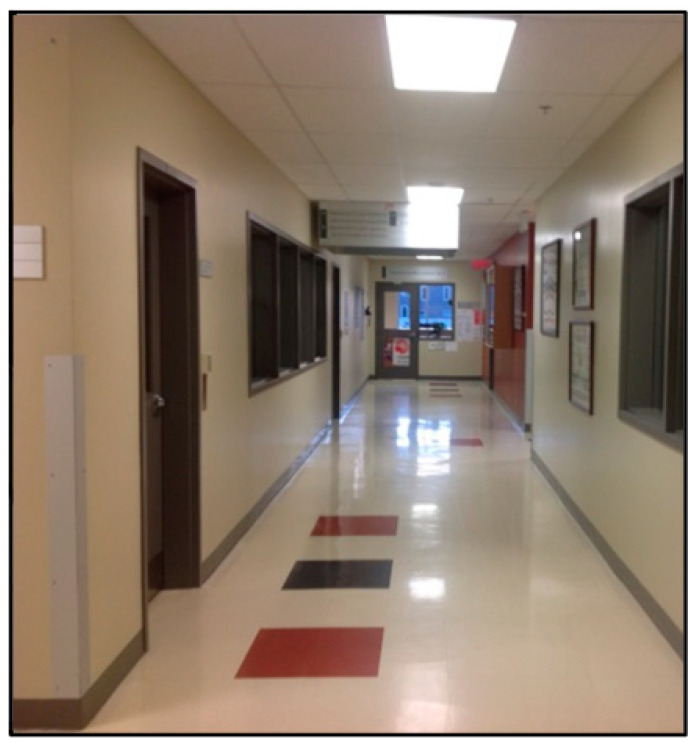
Image of real hallway at ÉTS.

**Figure 2 sensors-24-06307-f002:**
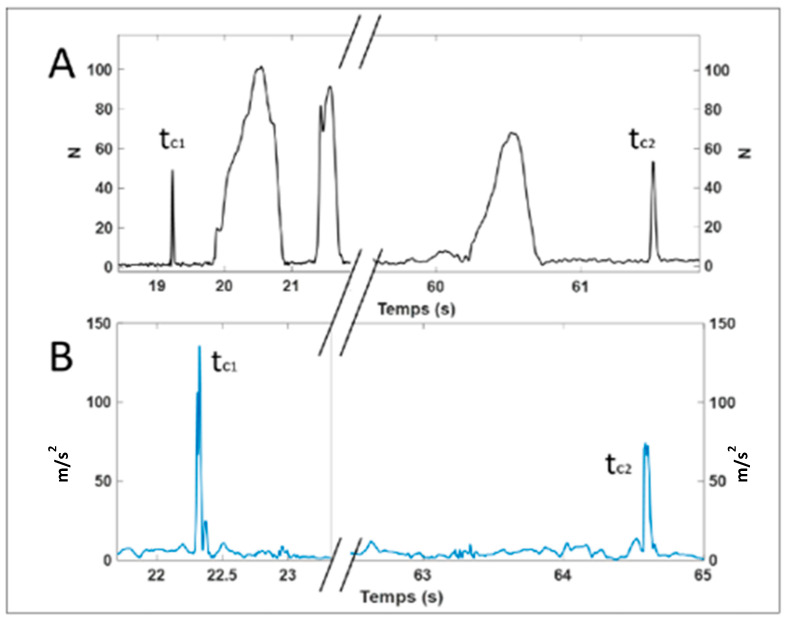
(**A**) Norm of the vector of reaction forces in (N); (**B**) norm of the vector of linear hand acceleration in (m/s^2^) during the 20 m propulsion. Tc1 and Tc2 correspond to the instant where the hand is kicking the handrim of the Smartwheel. These two events correspond to the beginning and the end of each trial.

**Figure 3 sensors-24-06307-f003:**
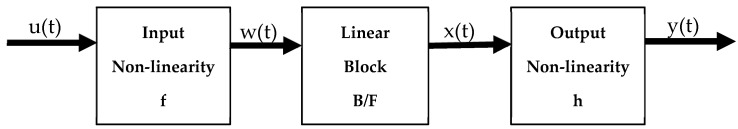
A SISO (single-input, single-output) H-W system which relates the input variable u(t) to the output variable y(t).

**Figure 4 sensors-24-06307-f004:**
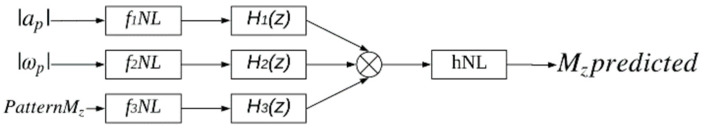
A general MISO H-W modeling approach.

**Figure 5 sensors-24-06307-f005:**
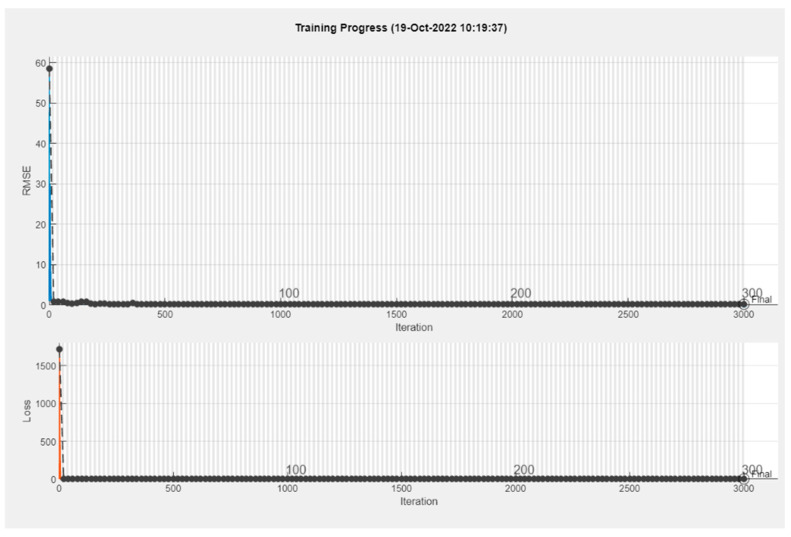
A successful BiLSTM training on MatLab.

**Figure 6 sensors-24-06307-f006:**
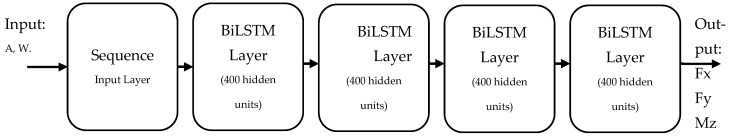
Our BiLSTM neural network model. The input corresponds to a vector of linear acceleration (A) and angular velocity vector (W). The output corresponds to the predicted forces (Fx, Fy) as well as the predicted moment (Mz).

**Figure 7 sensors-24-06307-f007:**
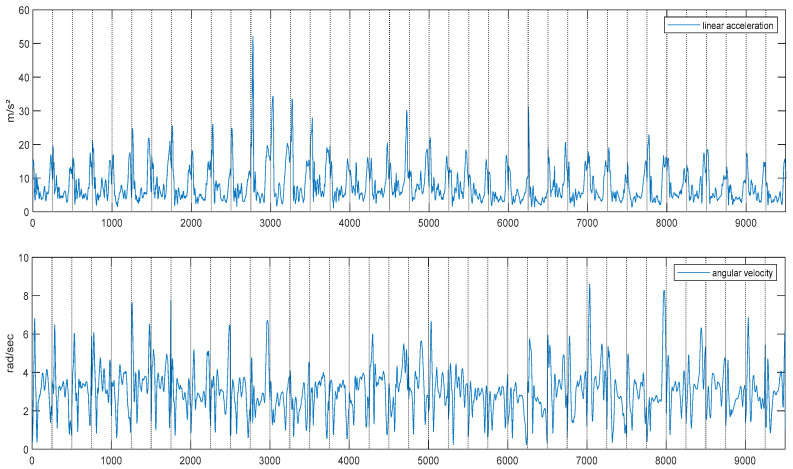
Typical input time-series data of linear acceleration (**Top**) and angular velocity (**Bottom**) during manual propulsion. The horizontal bar represents the frame and indicates 38 consecutives cycles of propulsions. The vertical bar indicates the beginning and end of the propulsion cycle period.

**Figure 8 sensors-24-06307-f008:**
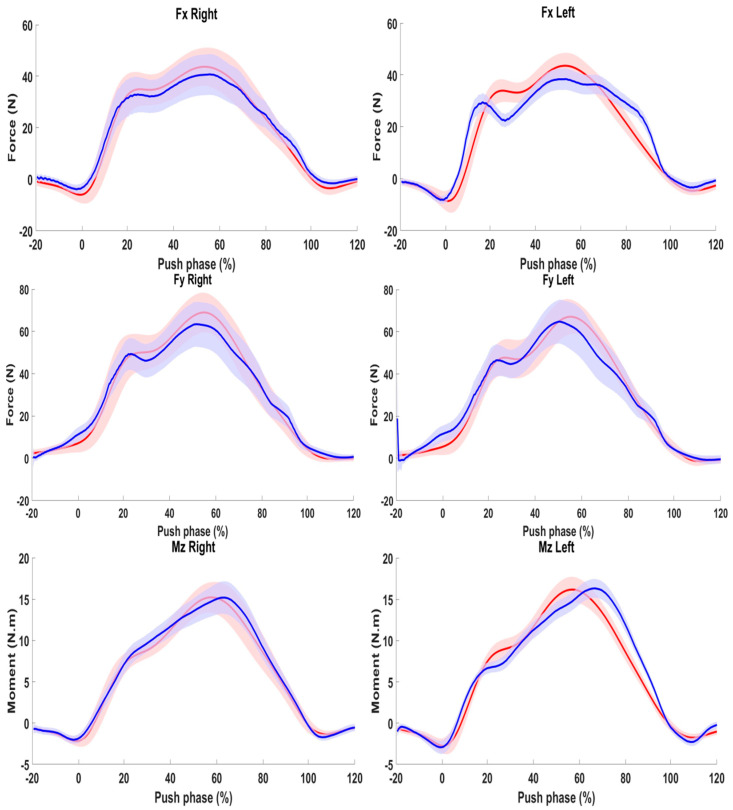
Prediction of horizontal forces Fx (**Top**), vertical forces Fy (**middle**), and sagittal moment (**Bottom**). Left column represents the right hand and the right column represents the left hand. Blue line represents the measured signal. Red Line represents predicted signal.

**Figure 9 sensors-24-06307-f009:**
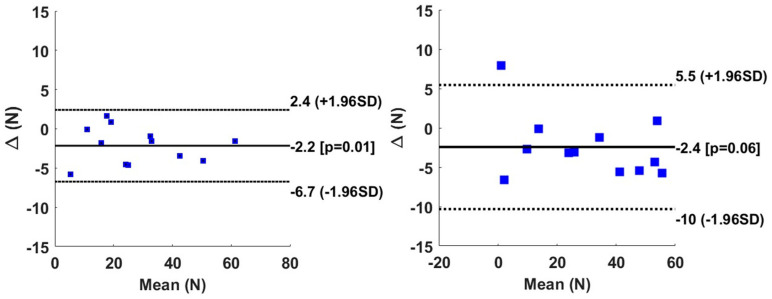
Bland–Altman graph of peak value for Fx for right hand (**left column**) and left hand (**right column**). Solid line represents the bias, whereas the dotted line represents the limit of agreement.

**Figure 10 sensors-24-06307-f010:**
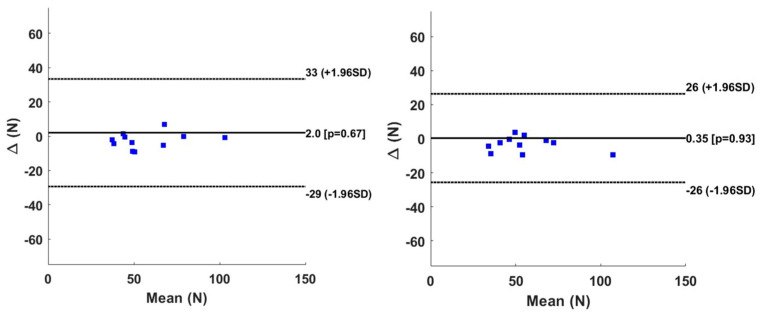
Bland–Altman graph of peak value for Fy for right hand (**left column**) and left hand (**right column**). Solid line represents the bias, whereas the dotted line represents the limit of agreement.

**Figure 11 sensors-24-06307-f011:**
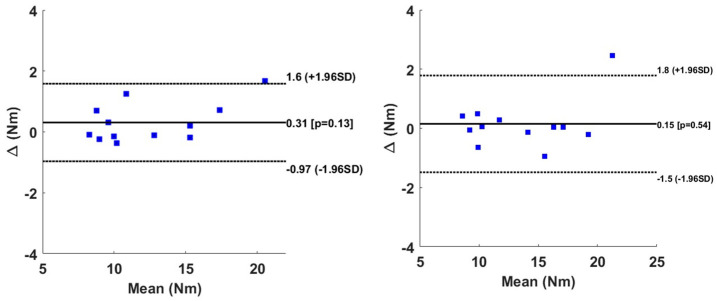
Bland–Altman graph of peak value for Mz for right hand (**left column**) and left hand (**right column**). Solid line represents the bias, whereas the dotted line represents the limit of agreement.

**Figure 12 sensors-24-06307-f012:**
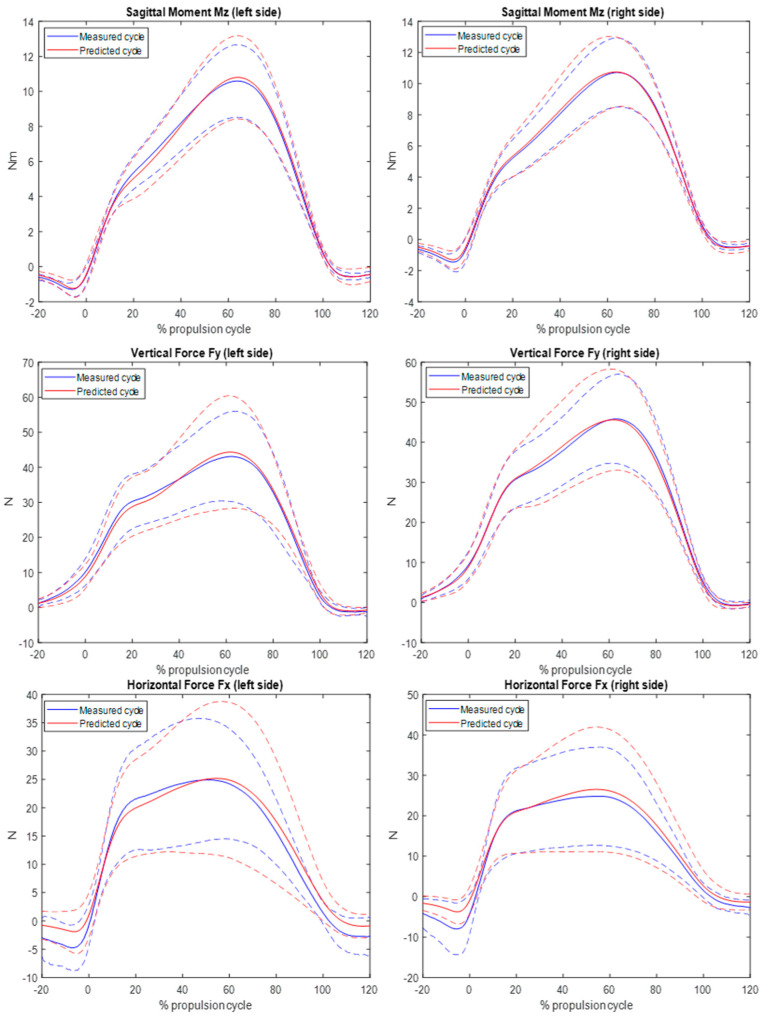
Prediction of horizontal and vertical forces and sagittal moments on the right and left sides using the BiLSTM approach. (The dotted line represents +/− the standard deviation). Blue line represents the measured signal. Red Line represents predicted signal.

**Table 1 sensors-24-06307-t001:** RMSE and MAE value for the Fx and Fy and Mz values throughout the propulsion cycle for the HW modeling approach.

Subject	Variable											
	Fx (N)				Fy (N)				Mz (N.m)			
	RMSE		MAE		RMSE		MAE		RMSE		MAE	
	Right	Left	Right	Left	Right	Left	Right	Left	Right	Left	Right	Left
1	4.5	4.7	3.5	3.8	9.5	6.8	7.5	5.5	1.2	0.8	0.9	0.7
2	5.0	3.7	4.3	2.8	8.9	8.6	6.9	6.7	1.1	0.8	0.8	0.6
3	4.4	3.8	3.2	2.9	7.4	10.4	5.8	8.1	1.1	1.0	0.8	0.8
4	5.0	4.1	4.1	3.0	9.6	7.7	7.8	6.2	0.8	1.1	0.6	0.9
5	7.4	6.9	5.7	5.3	9.5	11.3	7.2	9.1	1.6	1.4	1.4	1.1
6	4.7	6.1	3.8	5.0	10.5	11.9	8.5	8.8	1.6	1.4	1.3	1.2
7	3.7	6.3	3.0	1.3	5.6	7.9	4.5	6.1	0.9	1,6	0.7	1.3
8	6.7	4.3	5.3	3.3	7.2	6.6	5.8	5.4	1.3	1.3	1.0	1.1
9	6.1	7.0	5.0	5.4	10.7	13.8	8.6	10.7	2.2	2.7	1.8	2.2
10	7.9	6.5	5.9	5.2	15.7	13.7	12.7	10.9	1.7	1.8	1.4	1.5
11	5.8	9.7	4.6	8.3	9.4	8.1	7.7	6.5	1.3	1.2	1.0	0.9
Mean	5.6	5.7	4.4	4.2	9.4	9.7	7.5	7.6	1.3	1.3	1.0	1.1

**Table 2 sensors-24-06307-t002:** RMSE and MAE values for the Fx and Fy and Mz values throughout the propulsion cycle for the BiLSTM modeling approach.

Subject	Variable											
	Fx (N)				Fy (N)				Mz (N.m)			
	RMSE		MAE		RMSE		MAE		RMSE		MAE	
	Right	Left	Right	Left	Right	Left	Right	Left	Right	Left	Right	Left
1	4.4	6.1	3.6	5.1	7	5.8	5.6	4.6	1.2	1	0.9	0.8
2	6	4.5	5	3.8	6	6.4	4.6	4.9	0.9	0.9	0.7	0.7
3	8.9	3.7	7.3	3	5.6	7.1	4.2	5.3	0.9	1	0.7	0.8
4	5.6	5.3	4.8	4.2	9.6	7.7	7.6	5.9	0.9	1	0.7	0.8
5	12.3	12.7	9.7	10.9	7.1	7.2	5.4	5.7	1.6	1.5	1.2	1.2
6	7.2	10.3	5.9	8.7	6.9	9.2	5.5	7.4	1.5	1.5	1.2	1.2
7	7.3	6.6	6	5.5	8.2	9.1	6.4	7.2	1.3	1.5	1.1	1.1
8	4.8	5.5	3.9	4.6	5.1	5.4	4.1	4.3	0.9	0.9	0.7	0.7
9	14	5.7	11.2	4.6	10.1	10.9	8	8.8	1.4	1.5	1.1	1.2
10	5.6	9.1	4.7	7.7	7.8	6.5	6.2	5.2	1.3	1.4	1	1.1
11	5.9	11.4	4.6	9.2	8.8	8.2	6.7	6.4	1.4	1.3	1.1	1.1
Mean	7.4	7.3	6.1	6.1	7.5	7.6	5.8	5.9	1.2	1.2	0.99	0.99

## Data Availability

The data presented in this study are available on request from the corresponding author.

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
