# Peer review of "Handrim Reaction Force and Moment Assessment Using a Minimal IMU Configuration and Non-Linear Modeling Approach during Manual Wheelchair Propulsion"

_sensors, 2024, doi:10.3390/s24196307_

Round 1
Reviewer 1 Report
Comments and Suggestions for Authors
Overall comments
The paper introduces an innovative method for estimating handrim reaction forces and moments during wheelchair propulsion using a BiLSTM recurrent neural network.
The abstract lacks detailed examples of the results obtained which limits the reader’s understanding of the practical implications of the study.
The introduction is insufficient. It contains inaccuracies in method classification, lacks detailed results and context, and does not clearly compare existing approaches. This leads to confusion and a lack of clarity.
The Methods section requires further clarification to fully understand the implementation and how the proposed approach works in practice. Some parts of the methodology are not well justified and are sometimes inaccurate. Proper justification must be provided for this paper to be accepted.
The discussion is somewhat limited and needs more work to adequately address the implications of this work.
Abstract
The abstract is quite brief and would benefit from being expanded. It should include example results comparing the two methods, as well as a short conclusion that highlights the impact of the current proposed approach.
Introduction
---
Lines 40 to 56: This section needs to be rewritten for better cohesion. The comparison between the two main approaches (multibody and local segment models with (or without) machine learning) should be more clearly articulated and organized. Currently, the discussion of their strengths and weaknesses is somewhat scattered, which may leave the readers unclear about the practical implications of each approach. Each approach should be described in detail, followed by a clear statement of its advantages and disadvantages.
---
Since this study is mostly based on deep learning model, the introduction should talk more about other machine learning approaches that have been used in previous studies and their disadvantages compared to the current one.
---
The introduction also overlooks the rationale behind choosing a BiLSTM (Bidirectional Long Short-Term Memory) network over other models, such as traditional LSTMs or simpler recurrent neural networks. Providing this comparison would help clarify the specific advantages of using BiLSTM for predicting time series data related to handrim reaction forces.
---
A final point to consider is that using the Smartwheel, with its specific inertia characteristics and orientation, may introduce variability that affects the handrim reaction forces and moments. If the deep learning model is trained on data from the Smartwheel, it may not perform well with data from standard, non-instrumented wheels due to these differences. The Introduction lacks a discussion on how the model will handle this variability and ensure robust predictions across different types of wheels. It is essential to address how the model can be validated or adapted for general use beyond the Smartwheel to ensure its accuracy and reliability in various real-world scenarios.
---
The authors wrote: “A non-linear identification modelling approach and a recurrent neural network approach will be developed to estimate the magnitude and orientation of handrim reaction forces as well as the sagittal reaction moment during manual 81 wheelchair propulsion.”
Which approaches are considered? Specificity is required at the end of the introduction after justifying why these specific approaches are being used.
---
Materials and methods
Experimental set-up
---
The manuscript is missing details regarding the data pre-processing, which are important to validate and reproduce the results. No information is provided on whether preprocessing steps were applied to the sensor data.
Important preprocessing aspects to address would include:
1) gravity compensation: Was gravity removed from the accelerometer data?
2) Alignment Correction: Was any alignment correction performed to account for potential differences in sensor orientation between participants?
3) Filtering: Were the data filtered in any way (e.g., low-pass filtering) to remove noise or other artifacts?
4) Resampling: What resampling method was used?
5) Did you use the 6DOFS from IMU or the acceleration norm and the gyroscope norm? Providing clear detail about the tensor size used for training and testing would clarify the problem.
Some of these points are scattered around throughout the paper and should be gathered in one place. Any other preprocessing steps considered should be clearly stated in it.
---
Non-linear Hammerstein-Wiener modeling approach
---
Restructuring this part to first introduce the methods and then discuss the data organization specific to this approach would improve flow and clarity.
What are the parameters of the Hammerstein-Wiener model?
Related to my previous comments on IMU processing, you state in this part: “We choose deliberately to use the norm of hand linear acceleration and the norm of the hand angular velocity vector since these two time-series are invariant with respect to the orientation of the local coordinate system of each IMU. In this case, no alignment to anatomical reference system or calibration are necessary”.
Is the acceleration and gyroscope norm used only in this part and not in the Bi-LSTM part? If so, why?
Including a specific part about data processing will provide the reader with a comprehensive understanding of how data were prepared and processed.
---
Neural Network Long-Short Therm Memory (LSTM) approach
---
“Therm”?
Restructuring this part to first introduce the methods and then discuss the data organization specific to this approach would improve flow and clarity.
The authors wrote: “For this method, one subject was used as test subject each time (i.e leave-one out). The rest of the data was separated between training and validation data at 80-20% ratio. This separation is made to keep the maximum peak homogeneity between training and validation”.
If the goal is to develop a model that generalizes well to new participants, ensuring "maximum peak homogeneity" between training and validation data might not be ideal. Why was not a separate group of participants used for validation? This approach would allow the model to be trained until its performance starts to degrade with new participants, providing a better measure of generalization, especially since the mode is tested with one new participant.
Is the test procedure with one participant done 12 times? No information is provided about this.
The authors wrote: “When compared to Hammerstein-Wiener modeling approach, the BiLSTM model didn’t need a specific pattern as an input, and use only linear acceleration and angular velocity data as input.” Is the norm also used in this context?
What is the learning rate during training? Does it change during training, for example, through scheduling?
Did you perform any pre-processing, such as data normalization?
Why were 300 epochs chosen? Was an early stopping strategy used?
How is overfitting handled?
---
Results
Non-linear Hammerstein-Wiener modeling approach
---
Figure 8: Avoid using French in the figure (“Phase de poussees”)
Figure 9, 11: Where possible, extend the Y-axis scale to provide a clearer view of the results.
The results presented in Table 1 for each participant are interesting. To further enhance the analysis, it would be interesting to include a figure showing the root mean square error (RMSE) for each participant, comparing the predicted and actual values over time. It would be useful for checking for any trends or inconsistencies in the model's predictions.
---
Recurrent Neural Network: Bi-Long-Short Term Memory (BiLSTM) approach
---
Figure12: The legend is incomplete. What are the dot lines?
---
Discussion
---
Please add a limitation / future work part as well as a conclusion.
The "generic normalized pattern" does seem vague and could benefit from more detailed explanation. I couldn’t find a clear description of it.
The authors wrote: “High variability in wheelchair propulsion will have a better prediction with a non-linear transfer function whereas low variability is better predicted by the BiLSTM method.”
Neural networks are generally better suited for handling high variability and complex patterns, which is in contradiction with your statements.
---
Additionally, there is no mention of how the proposed methods will perform in real-world settings compared to controlled environments or how they might handle variability in wheelchair users' conditions.
---
Moreover, “non-linear transfer function” seems to refer to “Non-linear Hammerstein-Wiener modeling approach”. It is important to note a significant limitation of this method. Specifically, the approach relies on data from the first 15 cycles to make predictions. This means that it depends on existing data from a given participant to forecast new cycles. The paper does not demonstrate that this method works with data from new participants, which implies that using a Smartwheel might still be necessary at some point. This requirement can be a constraint and should be addressed clearly.
Given the current methodology, both modeling approaches have different purposes and potential applications. A more detailed discussion is needed to explore their relative advantages, limitations, and how they apply to different scenarios. Specifically, clarifying how each method handles variability and generalizes to new participants would strengthen the analysis. how each method manages variability and generalizes to new participants. This would provide a clearer understanding and enhance the overall analysis.
---
Others
The authors wrote: The study was conducted in accordance with the Declaration of Helsinki, and approved by Ethics Committee of École de technologie supérieure of Montreal (Canada)
Provide additional information, e.g., reference number and date of approval.
Comments on the Quality of English LanguageMinor editing of English language required
Author Response
Revision of the paper Reviewer 1:
Comments and Suggestions for Authors
Overall comments
The paper introduces an innovative method for estimating handrim reaction forces and moments during wheelchair propulsion using a BiLSTM recurrent neural network.
The abstract lacks detailed examples of the results obtained which limits the reader’s understanding of the practical implications of the study.
The introduction is insufficient. It contains inaccuracies in method classification, lacks detailed results and context, and does not clearly compare existing approaches. This leads to confusion and a lack of clarity.
The Methods section requires further clarification to fully understand the implementation and how the proposed approach works in practice. Some parts of the methodology are not well justified and are sometimes inaccurate. Proper justification must be provided for this paper to be accepted.
The discussion is somewhat limited and needs more work to adequately address the implications of this work.
Abstract
The abstract is quite brief and would benefit from being expanded. It should include example results comparing the two methods, as well as a short conclusion that highlights the impact of the current proposed approach.
Answer: The authors agree with the reviewer and the following new abstract has been added to replace the old one:
Abstract: Manual wheelchair propulsion represents a repetitive and constraining task, which leads mainly to the development joint injury in spinal-cord injured people. One of the main reasons is the load sustained by the shoulder joint during the propulsion cycle. Moreover, the load at the shoulder joint is highly correlated with the force and moment acting at the handrim level. The main objective of this study is related to the estimation of handrim reactions forces and moments during wheelchair propulsion using only a single inertial measurement unit per hand. Two approaches are proposed here: first, a method of identification of a non-linear transfer function based on Hammerstein Wiener (HW) modeling approach was used. The latter represents a typical multi-input single output in system engineering modelling approach. Secondly, a specific variant of recurrent neural network called BiLSTM is proposed to predicts time-series data of force and moments at the handrim level. Eleven subjects participated in the study during a linear propulsion protocol while the forces and moments were measured by a dynamic platform. The two input signals were the linear acceleration as well the angular velocity of wrist joint. The horizontal, vertical and sagittal moments were estimated by the two approaches. The mean average error (MAE) shows a value of 6.10 N and 4.30 N for the horizontal force for the BiLSTM and HW respectively. The results for the vertical direction show a MAE of 5.91 N and 7.59 N for BiLSTM and HW respectively. Finally, the MAE for sagittal moment varies from 0.96 Nm (BiLSTM) to 1.09 Nm for the HW model. The approaches seem similar with respect to the MAE and can be considered accurate knowing that the order of magnitude of the uncertainties of the dynamic platform was reported to be 2.2 N for horizontal and vertical forces, and 2.24 Nm for the sagittal moments. However, it should be noted that the HW necessitates the knowledge of the average force and patterns of each subject, whereas the BiLSTM method do not involve the average patterns which shows its superiority for time-series data prediction.
Introduction
---
Lines 40 to 56: This section needs to be rewritten for better cohesion. The comparison between the two main approaches (multibody and local segment models with (or without) machine learning) should be more clearly articulated and organized. Currently, the discussion of their strengths and weaknesses is somewhat scattered, which may leave the readers unclear about the practical implications of each approach. Each approach should be described in detail, followed by a clear statement of its advantages and disadvantages.
Answer: The authors agree with the reviewer and the following line has been added to replace the section from line 40 to 56 to the new paragraph form line 55 to 85:
The assessment of ground reaction forces during walking and running gait has been tackled for many years now, using inertial measurement unit (IMU). In fact, a systematic review has been reported spreading two decades for the estimation of ground reaction forces [8]. There are two major approaches to the problem: a multibody approach and local segment approach. The multibody approach is generally base on inverse dynamic as applied to the all-body segment i.e. the Newton-Euler formulation. The total vertical ground reaction forces are equal to the sum of mass of each segment multiply by the vertical linear acceleration of the segment minus the gravitation vector. This method is ideal for a perfect multibody model, but not in human model segments. The global approach suffers from three issues: the first one is related to skin tissue artefact of markers or sensors, the second one is related to the estimation of location and the mass of each segment of the body, the third one is more complex and related to the smooth transition in the double support phase. The study made by [20] reported an RMSE of almost 64 N for the vertical direction and 43 N for the horizontal direction as well as 18 Nm for the sagittal moment during normal walking. Recently, a probabilistic based on principal component analysis has been applied to the distribution of ground reaction forces between leg reduced the error on vertical ground reaction forces to 2.5 N/kg [21]. Moreover, when there is a multiple contact such as in lifting box the multi-body approach became inefficient and necessitates an optimization approach as well as a modeling of the contact between the environment and the human body: the latter can be handled by a complementary approach [22]. The latter estimated the vertical reaction forces with an RMSE of 0.51 N/kg The local approach is a method which try to relate a local information localized in one part of the body to the forces acting at the ground level. In [23], one accelerometer was fixed at the hip level, and a logarithmic regression equation was developed to predict peak ground reaction forces. The RMSE was large and almost close to 150 N. Recently The local approach based on three uniaxial load cells fixed on the shoes and combined with a deep learning method (Long Short-Term memory) was used to estimate the ground reaction forces [24]. The latter provide an RMSE of 65 N in vertical direction and 15 N in horizontal direction. The advantage of local approach is definitively the number of sensors used. Moreover, it seems that the closeness the sensors is to the contact zone the better are the estimated GRF.
Since this study is mostly based on deep learning model, the introduction should talk more about other machine learning approaches that have been used in previous studies and their disadvantages compared to the current one.
Answer::
We have a study that use LSTM to predict ground reaction force during gait. Artificial Neural Network is feed-forward network because the input is processed only in forward direction. Whereas recurrent neural network has a feed-back level however (RNN) are more complex. They save the output of processing nodes and feed the result back into the model (they did not pass the information in one direction only). This is how the model is said to learn to predict the outcome of a layer. Each node in the RNN model acts as a memory cell, continuing the computation and implementation of operations. RNN is well adapted to time-series prediction, and the variant named LSTM has an advantage of taken information in sequence and used them to predict future data in time-series. LSTM has the advantages to learn from times-series by a nonlinear cell input-output.
The introduction also overlooks the rationale behind choosing a BiLSTM (Bidirectional Long Short-Term Memory) network over other models, such as traditional LSTMs or simpler recurrent neural networks. Providing this comparison would help clarify the specific advantages of using BiLSTM for predicting time series data related to handrim reaction forces.
Answer:
Traditional forecasting of time-series has been handled for decades by the so-called ARIMA (autoregressive integrated moving average). These models perform reasonably well for short-term forecasts but their performance deteriorates severely for long-term predictions. Long Short-Term Memory (LSTM) was introduced to remember long input data and thus the relationship between the long input data and output is described in accordance with an additional dimension (e.g., time or spatial location). An LSTM network remembers long sequence of data through the utilization of several gates such as: input gate, forget gate, and output gate. The bidirectional LSTMs (BiLSTM) networks are a variation of normal LSTMs in which the desired model is trained not only from inputs to outputs, but also from outputs to inputs. In [24], the authors compared the performance of the ARIMA, LSTM and BiLSTM in forecasting 11 database time-series data. The results show the superiority of BiLSTM in terms of RMSE.
A final point to consider is that using the Smartwheel, with its specific inertia characteristics and orientation, may introduce variability that affects the handrim reaction forces and moments. If the deep learning model is trained on data from the Smartwheel, it may not perform well with data from standard, non-instrumented wheels due to these differences. The Introduction lacks a discussion on how the model will handle this variability and ensure robust predictions across different types of wheels. It is essential to address how the model can be validated or adapted for general use beyond the Smartwheel to ensure its accuracy and reliability in various real-world scenarios.
Answer:
Two unique Smartwheels have been designed originally before commercialisation (Cooper R.A, 1995). The original Smartwheel was a little bit heavy. Since the commercialisation two version have been designed with infra-red transmission and the last on in 2004 (Wifi transmission). The moment of inertia of the smartwheel is little bit higher than an ultra-light wheelchair. This is the version used in our study. In our teams and after discussion with one of our authors (FC), which works in adapted wheelchair sport activity, the role of inertia has an impact on the first push cycle, in the steady-state this impact is negligible. In this study we present the results of steady-state wheelchair propulsion. In this study we limit the use of Smartwheel with a zero-camber orientation with respect to the vertical plane. The non-zero camber will be addressed in future studies.
Cooper, R.A. Rehabilitation Engineering Applied to Mobility and Manipulation. (1995). IOP publishing Ltd. Chap 3. Biomechanics of Mobility and Manipulation, pp. 69-154.
The authors wrote: “A non-linear identification modelling approach and a recurrent neural network approach will be developed to estimate the magnitude and orientation of handrim reaction forces as well as the sagittal reaction moment during manual 81 wheelchair propulsion.”
Which approaches are considered? Specificity is required at the end of the introduction after justifying why these specific approaches are being used.
Answer:
The authors agree and the following text has been added from line 106 to 117.
The purpose of this study is to develop a new ambulatory local approach, which uses one inertial measurement unit (IMU) sensor at each wrist level. The IMU can provide two inputs: the 3D linear acceleration and 3D angular velocity vectors. Traditional forecasting of time-series data used a block-oriented identification method such as ARIMA [25] which represents a linear approach and performs poorly when there is a nonlinear relationship between input and output time-series. A non-linear identification modelling approach such as Hammerstein-Wiener (HW) exist and has been used in past to forecast time-series data and prove its superiority to the ARIMA, however it has never been used in wheelchair biomechanics [25]. The purpose of this study is to compare two approaches namely the HW and the BiLTSM in the forecasting of time-series of reaction forces and moment acting at the handrim level during manual wheelchair propulsion
Materials and methods
Experimental set-up
---
The manuscript is missing details regarding the data pre-processing, which are important to validate and reproduce the results. No information is provided on whether preprocessing steps were applied to the sensor data.
Important preprocessing aspects to address would include:
1) gravity compensation: Was gravity removed from the accelerometer data?
2) Alignment Correction: Was any alignment correction performed to account for potential differences in sensor orientation between participants?
3) Filtering: Were the data filtered in any way (e.g., low-pass filtering) to remove noise or other artifacts?
4) Resampling: What resampling method was used?
5) Did you use the 6DOFS from IMU or the acceleration norm and the gyroscope norm? Providing clear detail about the tensor size used for training and testing would clarify the problem.
Some of these points are scattered around throughout the paper and should be gathered in one place. Any other preprocessing steps considered should be clearly stated in it.
Answer: the authors thank the reviewer and provided the following text to describe the 5 elements as mentioned above. The new paragraph for line 127 to 143 has been modified:
The wheelchair was equipped with two Smartwheels which measured the handrim reaction forces on the anteroposterior and vertical directions, as well as the moment reaction on medial-lateral axis. The sampling frequency was fixed at 240 Hz. The Smartwheel software filtered the raw force and moment data with a 2nd order low-pass Butterworth filter with a cut-off frequency of 30 Hz. During the propulsion, the Xsens System (Mvn Biomech, Xsens Inc.) was used to model the head, trunk and upper-limb of the body which represent (10 Xsens IMU sensors). The calibration procedure in sitting position i.e. a T-pose has been made for each subject. The IMU data were originally processed by the Xsens system (i.e. internal filtering algorithm using Kalman filter). Moreover, gravity was removed by the Xsens system following the estimation of the quaternion of each sensors in the reference system of the Xsens. The Xsens has a biomechanical model which align the sensor information to the human body. In this study we use the only the norm of the linear acceleration and angular velocity vector for the left and right wrist joint. The sampling frequency of the Xsens was set at 120 Hz. A numerical synchronization was established by asking the subject to kick by his right hand the handrim at the beginning and the end of trial. These two events (Tc1 and Tc2) are distinguishable easily in the time-series of force and hand acceleration (Figure 2).
Non-linear Hammerstein-Wiener modeling approach
---
Restructuring this part to first introduce the methods and then discuss the data organization specific to this approach would improve flow and clarity.
What are the parameters of the Hammerstein-Wiener model?
Answer: We restructured the section used for Hammerstein-Wiener modeling approach. Indicating the numbers of parameters and reducing the text for the benefit of comprehension. The following text is now indicated from line 161 to line 200.
Related to my previous comments on IMU processing, you state in this part: “We choose deliberately to use the norm of hand linear acceleration and the norm of the hand angular velocity vector since these two time-series are invariant with respect to the orientation of the local coordinate system of each IMU. In this case, no alignment to anatomical reference system or calibration are necessary”.
Is the acceleration and gyroscope norm used only in this part and not in the Bi-LSTM part? If so, why?
Answer: for the BiLSTM the same time-series data have been used except that the generic pattern was not needed and only the norm of the linear acceleration and the norm of the angular velocity are used. It reduced then the input to only two time-series for each dependent variable i.e., Fx, Fy and Mz.
Including a specific part about data processing will provide the reader with a comprehensive understanding of how data were prepared and processed.
---
Neural Network Long-Short Therm Memory (LSTM) approach
---
“Therm”?
Answer : the correction has been made the word ‘therm’ was replaced by ‘term’
Restructuring this part to first introduce the methods and then discuss the data organization specific to this approach would improve flow and clarity.
The authors wrote: “For this method, one subject was used as test subject each time (i.e leave-one out). The rest of the data was separated between training and validation data at 80-20% ratio. This separation is made to keep the maximum peak homogeneity between training and validation”.
If the goal is to develop a model that generalizes well to new participants, ensuring "maximum peak homogeneity" between training and validation data might not be ideal. Why was not a separate group of participants used for validation? This approach would allow the model to be trained until its performance starts to degrade with new participants, providing a better measure of generalization, especially since the mode is tested with one new participant.
Answer : Yes, that was indeed the concern that came with taking validation subjects from among the 11 subjects. That's why we used a homogeneity approach. On a wider range of subjects, taking a separate group for validation could be more optimal than our current method.
Is the test procedure with one participant done 12 times? No information is provided about this.
Answer: Each subject was tested once on a network trained on the other subjects. We thus obtain 11 networks trained on different subjects and tested each time for the remaining one subject.
The authors wrote: “When compared to Hammerstein-Wiener modeling approach, the BiLSTM model didn’t need a specific pattern as an input, and use only linear acceleration and angular velocity data as input.” Is the norm also used in this context?
Answer: Yes, the time-series for the linear acceleration and the angular velocity are the same
What is the learning rate during training? Does it change during training, for example, through scheduling?
Answer : in ths first attempt, we did not study the learning rate, however during our use of the BiLSTM it did not vary. Future study with more data subject could help to analyse the influence of learning rate.
Did you perform any pre-processing, such as data normalization?
Answer: we normalize only the measured parameters it means the force and moment. We found out that dividing each cycle of time-series of force and moment with respect to the rms of the cycle gave us a high repetability. We conserve this rms to be able to regenerate the force in N and the moment in N.m.
Why were 300 epochs chosen? Was an early stopping strategy used?
Answer: The 300 epochs were chosen experimentally. They offered the best performance/speed ratio. No early stopping strategy was used.
How is overfitting handled?
Answer: yes, overfitting was monitored by validation data by a parameter which, after full training on all epochs, determines at which epochs the model is best trained without overfitting.
---
Results
Non-linear Hammerstein-Wiener modeling approach
---
Figure 8: Avoid using French in the figure (“Phase de poussees”)
Answer: Figure 8 is written in English now
Figure 9, 11: Where possible, extend the Y-axis scale to provide a clearer view of the results.
Answer: Figure 9-11 have now a clear representation
The results presented in Table 1 for each participant are interesting. To further enhance the analysis, it would be interesting to include a figure showing the root mean square error (RMSE) for each participant, comparing the predicted and actual values over time. It would be useful for checking for any trends or inconsistencies in the model's predictions.
Answer: The authors agree with the reviewers, and Table-1 provides now the same information as for BiLSTM i.e. the RMSE and the MAE for each subject throughout the continuous cycle of propulsion. It is now possible to compare the final results form HW and BiLSTM approaches.
---
Recurrent Neural Network: Bi-Long-Short Term Memory (BiLSTM) approach
---
Figure12: The legend is incomplete. What are the dot lines?
Answer: The dot line represents the +/- standard deviation, it has been added to the figure legend
---
Discussion
---
Please add a limitation / future work part as well as a conclusion.
Answer: a paragraph has been added at the end of the discussion part for limitation and future work.
This study however is limited to straightforward propulsion i.e., in linear direction, we did not assess a curvilinear or slope path. Moreover, the wheel camber in this study was fixed to zero i.e. the inplane of the smartwheel are the vertical plane. In many wheelchairs as those dedicated for spinal-cord injury and for sport activity the camber is important. The subjects that participate in this study are young and healthy i.e., non-wheelchair users [19]. The generalization of this method to wheelchair users such as spinal-cord injury should be done in future study. Another aspect about the normalization used in this study using the rms of the cycle as a base and the use of this base explicitly for the HW method is considered as a limitation. The generic pattern is also a limitation of the HW method. Future work should be done to understand how to deformalize the time-series signal after the prediction process.
The "generic normalized pattern" does seem vague and could benefit from more detailed explanation. I couldn’t find a clear description of it.
Answer: The generic pattern is obtained after normalisation of force or moment signal by their rms value for each propulsion cycle. Afterwards, the time-normalisation ie. Form -20% to 120% of the time all the cycles were averaged. We then obtain what we call a generic normalize pattern. This generic pattern is the input in the HW modeling approach and if the subject had 38 cycles of propulsions the input time-series will be a 38 replicates of the signal. The process in HW serve in a sort of a modulation of this pattern using actual linear acceleration and angular velocity of the wrist jointé
The authors wrote: “High variability in wheelchair propulsion will have a better prediction with a non-linear transfer function whereas low variability is better predicted by the BiLSTM method.”
Neural networks are generally better suited for handling high variability and complex patterns, which is in contradiction with your statements.
Answer : the authors agrees with the reviewers with the fact that neural network handle the complexity i.e. a strong non-linearities. However, our variability here concerns the amplitude of signal i.e. the normalization process which in neural network need to have a variable with zero-mean and one for standard deviation. In our cases we normalize with the rms of the signal cycle to be able to use the same type of signal as the HW modelling approach. When we use this type of normalization the BiLSTM lead sometimes to a rather less prediction accuracy. However, on the average the BiLSTM is more powerfull than the HW. We decide than to remove the sentences from the paper.
---
Additionally, there is no mention of how the proposed methods will perform in real-world settings compared to controlled environments or how they might handle variability in wheelchair users' conditions.
Answer: We test our method by collecting data in indoor situation a corridor at the École de technologie supérieure with one condition of ground friction. In others situation i.e. carpet or concrete or woods the handrim reaction forces changes as well as the population. We have indicated in a limitation paragraph the issue about spinal-cord injured person.
---
Moreover, “non-linear transfer function” seems to refer to “Non-linear Hammerstein-Wiener modeling approach”. It is important to note a significant limitation of this method. Specifically, the approach relies on data from the first 15 cycles to make predictions. This means that it depends on existing data from a given participant to forecast new cycles. The paper does not demonstrate that this method works with data from new participants, which implies that using a Smartwheel might still be necessary at some point. This requirement can be a constraint and should be addressed clearly.
Given the current methodology, both modeling approaches have different purposes and potential applications. A more detailed discussion is needed to explore their relative advantages, limitations, and how they apply to different scenarios. Specifically, clarifying how each method handles variability and generalizes to new participants would strengthen the analysis. how each method manages variability and generalizes to new participants. This would provide a clearer understanding and enhance the overall analysis.
Answer : We agree with the reviewers that the HW necessitates the knowledge with a generic pattern which is obtained by the smartwheel originally. We made in a limitation of this study this argument. It should be reminded that this study is considered a preliminary study concerning the possibilities to by -pass the use of the SmartWheel. It is definitely clear that BiLSTM will operate better than HW if we omit the generic pattern of the force or the moment. Further studies will tackle this problem i.e. the one concerning a new participant, having a big database we can construct a more reliable neural network that span the subject variability.
---
Others
The authors wrote: The study was conducted in accordance with the Declaration of Helsinki, and approved by Ethics Committee of École de technologie supérieure of Montreal (Canada)
Provide additional information, e.g., reference number and date of approval.
Answer: This project has been approved by the Ethics committee of the École de technologie supérieure de Montréal in Winter 2015 where the data has been collected originally the file number is H20150508.
Comments on the Quality of English Language
Minor editing of English language required
Answer: English has been verified
Submission Date
30 July 2024
Date of this review
29 Aug 2024 03:26:33
Bottom of Form
© 1996-2024 MDPI (Basel, Switzerland) unless otherwise stated
Reviewer-2
Open Review
(x) I would not like to sign my review report
( ) I would like to sign my review report
Quality of English Language
( ) I am not qualified to assess the quality of English in this paper.
( ) The English is very difficult to understand/incomprehensible.
( ) Extensive editing of English language required.
( ) Moderate editing of English language required.
(x) Minor editing of English language required.
( ) English language fine. No issues detected.
|
Yes |
Can be improved |
Must be improved |
Not applicable |
|
|
Does the introduction provide sufficient background and include all relevant references? |
(x) |
( ) |
( ) |
( ) |
|
Is the research design appropriate? |
(x) |
( ) |
( ) |
( ) |
|
Are the methods adequately described? |
( ) |
(x) |
( ) |
( ) |
|
Are the results clearly presented? |
(x) |
( ) |
( ) |
( ) |
|
Are the conclusions supported by the results? |
(x) |
( ) |
( ) |
( ) |
Comments and Suggestions for Authors
- 1. Please reduce the number of hyphenations in the text for easily reading.
- 2. The configuration of the computer used in LINE 165 may be incorrect. What if 32,0 Go RAM?
- The representation of decimal points is confusing, with periods used in some places and commas used in most places. Please use the internationally accepted representation, that is, use a period as the decimal point representation.
- Something in Figure 8 is cut and disappear. Please be careful about the size of graphs.
- Table 2 and Table 3 are lost. And there is no caption for Table 4. These mistakes are very serious. Please correct them.
- There are two numbers in each cell of Table 1. But I cannot find any description about them. What is the meaning of the upper number and what is the meaning of the lower number (inside parentheses)?
Comments on the Quality of English Language
There are some typing errors and a few grammatical errors.
Submission Date
30 July 2024
Date of this review
22 Aug 2024 16:35:36
Reviewer 2 Report
Comments and Suggestions for Authors
1. Please reduce the number of hyphenations in the text for easily reading.
2. The configuration of the computer used in LINE 165 may be incorrect. What if 32,0 Go RAM?
3. The representation of decimal points is confusing, with periods used in some places and commas used in most places. Please use the internationally accepted representation, that is, use a period as the decimal point representation.
4. Something in Figure 8 is cut and disappear. Please be careful about the size of graphs.
5. Table 2 and Table 3 are lost. And there is no caption for Table 4. These mistakes are very serious. Please correct them.
6. There are two numbers in each cell of Table 1. But I cannot find any description about them. What is the meaning of the upper number and what is the meaning of the lower number (inside parentheses)?
Comments on the Quality of English Language
There are some typing errors and a few grammatical errors.
Author Response
Response to reviewers 2
- The hyphenation is imposed by the gabarit of sensors from. We try to minimize that as long as we could.
- The 32 Go means giga-octet of memory, however in the last version we remove the sentences.
- The decimal point has been corrected for all tables an figures
- Figure has been modified
- The text has been changes and there is only 2 tables now in the manuscript
- Table 1 has changed and now there is only one line for the RMSE and the MAE which looks similar to the BiLSTM modeling approach
Round 2
Reviewer 1 Report
Comments and Suggestions for Authors
The authors have largely addressed the previous comments, but there are still a few minor modifications needed for clarity.
Abstract
Line 34 – 35: “shrovided in this study show the possibility to measure dynamic forces acting at handrim during wheelchair manual propulsion in ecological environments.”
This part of the text seems detached from the abstract. Please correct it
Materials and methods
Neural Network Long-Short Therm Memory (LSTM) approach
Line 222 – 223: “The “Sequence Input Layer” adapts input data for 3 BiLSTM of 400 hidden layers followed by a fully connected Layer to predict our data”.
I believe the authors mean 400 hidden "units" rather than "layers." Please correct it
Others
The authors answer to my previous comments: “This project has been approved by the Ethics committee of the École de technologie supérieure de Montréal in Winter 2015 where the data has been collected originally the file number is H20150508.”
While the authors provided the ethical approval details in their response, this information is still not included in the manuscript. Please ensure that it is added to the appropriate section.
Author Response
Line 34 – 35: “shrovided in this study show the possibility to measure dynamic forces acting at handrim during wheelchair manual propulsion in ecological environments.”
This part of the text seems detached from the abstract. Please correct it
Answer
The following sentence has been now modified and added at the end of the abstract.
The results provided in this study show the possibility to measure dynamic forces acting at handrim during wheelchair manual propulsion in ecological environments.
Line 222 – 223: “The “Sequence Input Layer” adapts input data for 3 BiLSTM of 400 hidden layers followed by a fully connected Layer to predict our data”.
I believe the authors mean 400 hidden "units" rather than "layers." Please correct it
Answer:
The text has been corrected and it specify 400 hidden units (line 223). Consequently figure 6 is modified including the specification of 400 hidden units.
The authors answer to my previous comments: “This project has been approved by the Ethics committee of the École de technologie supérieure de Montréal in Winter 2015 where the data has been collected originally the file number is H20150508.”
While the authors provided the ethical approval details in their response, this information is still not included in the manuscript. Please ensure that it is added to the appropriate section.
Answer :
The reference number for the ethic committee has been added at line 124